# Prevalence and factors associated with Intestinal parasitosis among adults with diabetes mellitus attending a tertiary care facility in Northern Uganda: A hospital-based cross-sectional study

Hillary Ngolobe[1]*, Martin Kugonza[1], Phillip Nkwatsibwe[1], Anthony Kato[1], Donald Otika[1], Morrish Obol Okello[1], Daniel S. Ebbs[2], Felix Bongomin[1]*

1 Faculty of Medicine, Gulu University, Gulu City, Uganda, 2 Section of Paediatric Critical Care, Yale University School of Medicine, New Haven, Connecticut, United States of America

* hillaryngolobe1@gmail.com (HN); drbongomin@gmail.com (FB)

## Abstract

Intestinal parasitosis (IP) constitutes a significant public health problem in developing countries, where more than three billion people worldwide are infected with one or more intestinal parasites. Diabetes mellitus (DM) is associated with an increased risk of acquiring IP. This study aimed to determine the prevalence and associated factors of IP among patients with DM attending Gulu Regional Referral Hospital (GRRH) in Northern Uganda. A hospital-based cross-sectional study was conducted using a semi-structured questionnaire and stool sample analysis. Descriptive statistics were used to summarize patient characteristics, and bivariate logistic regression was performed to assess associated factors of IP among DM patients attending GRRH. Out of the 267 eligible patients, 201 participated, indicating a response rate of 75.3%. The majority were female (68.2%, n = 137), with a mean age of 55.8 ± 12.8 years. Almost all participants had type 2 diabetes mellitus (99.0%, n = 199), with a mean duration of 7.0 ± 6.8 years and a mean fasting blood glucose of 11.9 ± 6.4 mmol/L. The prevalence of intestinal parasitosis was 19.4% (39/201). *Entamoeba coli*, a non-pathogenic commensal, accounted for 35 cases (89.7%), while the four pathogenic parasites; *Hymenolepis nana, Giardia duodenalis, Entamoeba histolytica,* and *Iodamoeba buetschlii,* were each identified in one case (2.6%). Patients who did not seek medical attention for the abdominal symptoms were three times more likely to have intestinal parasitoses (cOR =3.0, 95% CI: 1.18–7.53, P = 0.014). The notable prevalence of IP among patients with DM suggests heightened susceptibility in this population, which may adversely affect their quality of life. Routine screening for intestinal parasites is therefore recommended for diabetic patients.

**Data availability statement:** All relevant data are within the paper and its Supporting Information files.

**Funding:** The author(s) received no specific funding for this work.

**Competing interests:** The authors have declared that no competing interests exist.

## 1. Introduction

Intestinal parasitosis (IP) constitutes a critical problem of public health importance, especially in developing countries. It is estimated that globally, approximately 3.5 billion people are infected by various types of IPs, and over 200,000 deaths are reported annually [1]. It has been highlighted that IPs are one of the major causes of digestive disorders (such as diarrhea, nausea, and vomiting), chronic malabsorption and malnutrition, and failure to thrive, especially among high-risk groups, namely, children, pregnant women, and immunocompromised patients [2–4]. Personal hygiene, environmental sanitation, level of education, occupation, and overcrowding are probable factors that predispose individuals to intestinal parasitic infections [5]

Diabetes Mellitus (DM) is a chronic metabolic disease affecting over 589 million adults in 2024 and expected to rise to 853 million by 2050 [6]. DM is characterized by chronic hyperglycemia and is due to either deficient insulin release or reduced response to insulin [7]. DM increases the risk of infections, and in particular intestinal parasites, due to impaired innate and adaptive immune responses [8]. A recent systematic review found that intestinal parasites were more prevalent in patients with DM compared to those without [9], and therefore stands out as a risk factor for intestinal parasites

The global prevalence of intestinal parasites among patients with DM has been estimated to be 24.4% [9] with 31% in Africa [10]. Country-specific studies have reported a prevalence of 19.5% in Ethiopia [11], 12.5% in Ghana [12] and 18.7% in Nigeria [13]. Uganda's prevalence of IP among patients with diabetes mellitus has not been studied yet.

Most studies conducted in Uganda on intestinal parasitosis have focused on children, considering them as at higher risk than the other population groups [14,15]. However, a previous study conducted in Ethiopia highlighted that adults who are immunocompromised are also at risk of getting intestinal parasitic infections, underscoring the need for more data to be obtained among these populations [16]. Our study is timely to add to the literature on intestinal parasitosis in Uganda, particularly among the adult population that is immunocompromised.. Therefore, our study aims to assess the prevalence and associated factors of IP among DM patients attending Gulu Regional Referral Hospital (GRRH) in Northern Uganda.

## 2. Methods

### 2.1. Study design and setting

This was a hospital-based cross-sectional study utilizing a quantitative approach, conducted from the 28th of October 2024 to the 3rd of March 2025. This study was designed in accordance with the STROBE guidelines for cross-sectional studies. It was carried out at the DM clinic of GRRH, which operates every Monday with an attendance of over 50 patients.

### 2.2. Study population, Inclusion and exclusion criteria

Adults receiving care from the diabetic clinic at GRRH were targeted. Patients aged 18 years and above, with a physician-confirmed diagnosis of DM, and who provided

a written informed consent were included. Patients who received anti-helminth drugs in the past 2 weeks, those with hyperglycemic crisis, and those who had constipation were excluded.

## 2.3. Sample size determination and sampling procedure

The sample size (n) was calculated using Cochran's formula $n = Z^2 \cdot P(1-P)/d^2$: considering z = 1.96 (for a 95% confidence interval), p = estimated prevalence, 19.5% [11], and d = 0.05 (margin of error at 5%)an expected response rate of 90%, the sample size was 267. A convenient sampling technique was employed, and eligible participants were chosen consecutively. This sampling technique was used because of the difficulty in accessing study participants and the limited time available for conducting data collection.

## 2.4. Research instrument and Data collection

An interviewer-administered semi-structured questionnaire was used to collect data on socio-demographics, as well as factors associated with STHI. The questionnaire was adapted based on a previous study on intestinal parasites among patients with DM [11]. Participants were interviewed after consultation with the physician when they came for routine care at the DM clinic of GRRH. The interviews were conducted in the clinical room provided by the hospital to ensure confidentiality. A stool sample was then obtained from each of the participants and taken for analysis within 4 hours of sample collection. Two [2] thick smears were prepared for analysis using both the Kato Katz and Odongo-Aginya stains. The stool analysis was performed by 1 laboratory technician and a professor of parasitology.

## 2.5. Data management and analysis

Completed questionnaires were securely stored and then entered into Excel before being exported to Stata version 18 for analysis. Descriptive statistics were performed, and categorical variables were summarized as frequencies and percentages. Numerical data were assessed for normality using the Shapiro-Wilk test and then summarized as means and standard deviations or median and interquartile range, accordingly.

## 2.6. Ethical considerations

Ethical approval was obtained from Gulu University Research Ethics Committee (GUREC, GUREC-2024–926). Initial approval was granted on 29/09/2024, and data collection commenced on 28/10/2024. Amendments were made, and the approval for the amendments was on 30/05/2025Administrative clearance was obtained from GRRH Institutional Review Board.. Written informed consent was obtained, and the purpose of the study was explained to the study participants. The participants were informed of their right to withdraw from the study at any time. All data collected was kept confidential and anonymous. All the other ethical principles outlined in the Declaration of Helsinki were observed.

# 3. Results

## 3.1. Study enrolment

A total of 252 patients were screened to participate in the study. Of these, 51 were excluded. Those excluded were due to: constipation and could therefore not provide a stool sample (29), recent use of anthelmintics and therefore there is a possibility of eradicating the helminths [20], and those in a hyperglycemic crisis and were therefore unstable to participate in the study [2]. Eventually, 201 participants participated in the study, indicating a 75.3% (n = 201/267) response rate **Fig 1**

## 3.2. Sociodemographic characteristics of people with DM attending GRRH

The mean age of participants was 55.8 ± 12.8 years, with a majority being female (n = 137, 68.2%). Most were of Acholi ethnicity (n = 178, 88.6%) and married (n = 143, 71.1%). Educational attainment was low, with over half having

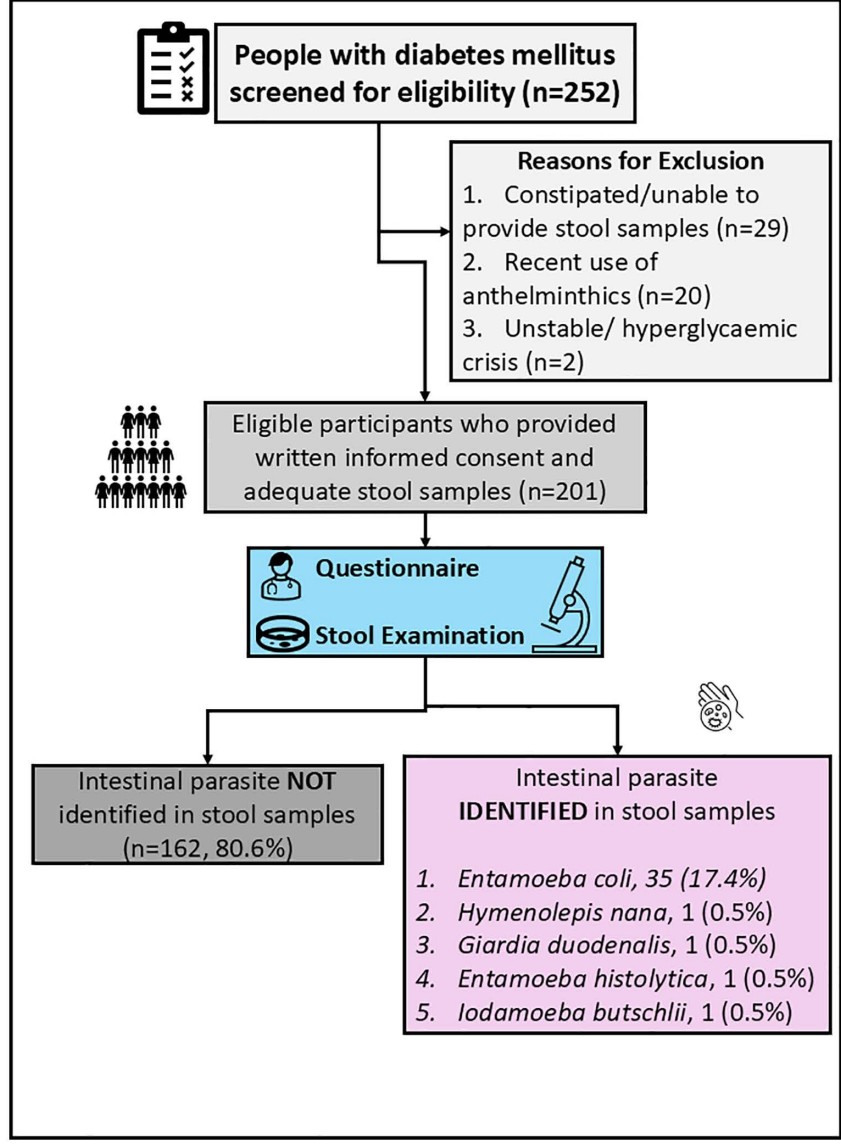

**Fig 1. Study enrolment diagram.**

only primary education (n = 106, 52.7%) and 20.9% (n = 42) having no formal education. Slightly more participants resided in urban areas (n = 105, 52.2%) than rural areas (n = 96, 47.8%). The majority were self-employed (n = 182, 90.5%). Christianity was the dominant religion (n = 197, 98.0%), with Catholics forming the largest denomination (n = 114, 56.7%), Table 1

### 3.3. Clinical characteristics of people with DM attending GRRH

Almost all participants had type 2 diabetes (n = 199, 99.0%), with a mean duration of 7.0 ± 6.8 years and a mean fasting blood sugar of 11.9 ± 6.4 mmol/L. Glycaemic control was poor, with most having uncontrolled fasting plasma glucose (n = 148, 73.6%). Diabetes-related complications were reported in 86.6% (n = 174), predominantly neuropathy (n = 156,

**Table 1. Socio-demographic characteristics of people with DM attending GRRH.**

| Variable (N=201) | Frequency | Percent |
|---|---|---|
| Age, mean ± SD, years | 55.8 | 12.8 |
| Gender | | |
| Female | 137 | 68.2 |
| Male | 64 | 31.8 |
| Tribe | | |
| Acholi | 178 | 88.6 |
| Others* | 23 | 11.4 |
| Marital status | | |
| Married | 143 | 71.1 |
| Widowed | 42 | 20.9 |
| Divorced | 10 | 5.0 |
| Single | 6 | 3.0 |
| Education level | | |
| No formal education | 42 | 20.9 |
| Primary | 106 | 52.7 |
| Secondary | 36 | 17.9 |
| Tertiary | 17 | 8.5 |
| Residence | | |
| Rural | 96 | 47.8 |
| Urban | 105 | 52.2 |
| Occupation | | |
| Civil servants | 7 | 3.5 |
| Self-employed | 182 | 90.5 |
| Unemployed | 12 | 6.0 |
| Religion | | |
| Christian | 197 | 98.0 |
| Catholic | 114 | 56.7 |
| Born again | 43 | 21. 4 |
| Anglican | 39 | 19.4 |
| SDA | 1 | 0.5 |
| Moslem | 4 | 2.0 |

*Other tribes: Langi (13, 6.5%), Alur (5, 2.5%), Lugbara, Muganda, Mukiga, munyankole, munyoro (1 each, 0.5%)

77.6%) and retinopathy (n = 100, 49.8%). Most participants were on oral anti-hyperglycaemic medications alone (n = 172, 85.6%). Co-morbidities were present in 73.1% (n = 147), primarily hypertension (n = 141, 70.1%), Table 2.

### 3.4. Risk factors for intestinal parasitosis infection among people with DM attending GRRH

A family history of intestinal parasitosis was reported by 12.9% (n = 26). Most participants often used footwear out-doors (n = 149, 74.1%) and practiced regular handwashing before eating (n = 195, 97.0%). Pet ownership was common (n = 136, 67.7%). The main sources of water were boreholes (n = 117, 58.2%) and natural sources (n = 42, 20.9%), while most used pit latrines for faecal disposal (n = 178, 88.6%). A high proportion reported current gastrointestinal symptoms (n = 146, 72.6%), predominantly abdominal pain (n = 120, 59.7%) and gastrointestinal upset (n = 93, 46.3%),

**Table 2. Clinical characteristics of people with DM attending GRRH.**

| Variable, N=201 | Frequency | Percent |
|---|---|---|
| Type of diabetes | | |
| Type 1 | 2 | 1.0 |
| Type 2 | 199 | 99.0 |
| Duration of diabetes, mean ± SD; years | 7.0 | 6.8 |
| Fasting blood sugar, mean ± SD; mmol/L | 11.9 | 6.4 |
| Glycaemic control | | |
| Controlled | 53 | 26.4 |
| Uncontrolled | 148 | 73.6 |
| Diabetes complications | | |
| Yes | 174 | 86.6 |
| No | 27 | 13.4 |
| Complications of diabetes (n=174) | | |
| Neuropathy | 156 | 77.6 |
| Retinopathy | 100 | 49.8 |
| Nephropathy | 9 | 4.5 |
| Current anti-hyperglycaemic therapy | | |
| All oral medicines | 172 | 85.6 |
| Insulin and oral medicines | 17 | 8.5 |
| Insulin alone | 12 | 6.0 |
| Co-morbidities | | |
| Yes | 147 | 73.1 |
| No | 54 | 26.9 |
| Proportions with co-morbidities (n=147) | | |
| Hypertension | 141 | 70.1 |
| HIV | 6 | 3.0 |
| Peptic ulcer disease | 3 | 1.5 |
| Viral hepatitis | 2 | 1.0 |
| Chronic heart disease | 1 | 0.5 |

though only 31.8% (n = 64) had sought physician consultation, with few symptoms linked to intestinal parasitosis (n = 4, 2.0%), Table 3.

### 3.5. Prevalence of intestinal parasitosis among people with DM attending GRRH

Of the 201 participants, 39 (19.4%) were found to have intestinal parasitosis. *Entamoeba coli*, a non-pathogenic commensal, constituted the majority (35 cases, 89.7%), while the four pathogenic parasites-. *Hymenolepis nana, Giardia duodenalis, Entamoeba histolytica,* and *Iodamoeba buetschlii* were each identified in one case (2.6%). Figs 2 and 3

### 3.6. Factors associated with intestinal parasitosis among patients with DM attending GRRH

We performed simple logistic regression at bivariate analysis and a p < 0.05 was considered statistically significant. Only one factor was found to be statistically significant; patients who did not seek medical attention for the abdominal symptoms were three (3) times more likely to have intestinal parasites (cOR =3.0, 95 CI: 1.18–7.53, P = 0.014.) Table 4.

**Table 3. Risk factors for intestinal parasitosis among people with DM attending GRRH.**

| Variable, N=201 | Frequency | Percent |
|---|---|---|
| Family history of intestinal parasitosis, yes | 26 | 12.9 |
| Use of footwear while walking outdoors | | |
| Never | 2 | 1.0 |
| Rarely | 50 | 24.9 |
| Often | 149 | 74.1 |
| Pet animals at home, yes | 136 | 67.7 |
| Handwashing before eating | | |
| Rarely | 1 | 0.5 |
| Sometimes | 5 | 2.5 |
| Always | 195 | 97.0 |
| Water source | | |
| Bore hole | 117 | 58.2 |
| Natural water sources | 42 | 20.9 |
| Pipped water | 32 | 15.9 |
| Mixed sources | 10 | 5.0 |
| Human faecal disposal | | |
| Pit latrines | 178 | 88.6 |
| Toilet | 23 | 11.4 |
| Current gastrointestinal symptoms | | |
| Symptomatic | 146 | 72.6 |
| Asymptomatic | 55 | 27.4 |
| Proportion of symptoms, n=146 | | |
| Abdominal pain | 120 | 59.7 |
| Nausea, vomiting and diarrhoea | 93 | 46.3 |
| Weightless | 26 | 12.9 |
| Physician consultation for gastrointestinal symptom, n=146; yes | 64 | 31.8 |
| Symptom linked to IP | 4 | 2.0 |

## 4. Discussion

In this study, we aimed to determine the prevalence and factors associated with IP among adults with DM attending GRRH. The prevalence of intestinal parasitosis was 19.4%. *Entamoeba coli*, a non-pathogenic commensal, constituted the majority (89.7%), while the four pathogenic parasites. *Hymenolepis nana, Giardia duodenalis, Entamoeba histolytica, and Iodamoeba buetschlii* were each identified in one case (2.6%). The presence of Entamoeba coli among the participants indicates fecal-oral contamination and poor sanitation among the participants, increasing the risk of exposure to truly pathogenic parasites. This calls for the need to improve the WASH (Water, Sanitation, and Hygiene) strategy in the communities to reduce the risk of fecal-oral contamination.

Our findings are consistent with prior studies, which found the prevalence of intestinal parasites among DM patients at 19.5% in Ethiopia [11], 20.6% in Sudan [17] and 18.7% in Nigeria [13]. However, our study reported a higher prevalence than what was found in Thailand and India, which were 11.31% and 13.6% respectively. [18,19]. The differences in the findings could be explained by the differences in the characteristics of the study populations. The studies with consistent findings were all conducted in sub-Saharan Africa, which is among the regions recognized for having the highest prevalence for IP, hence moreless similar study population. Interestingly, the other studies that reported lower prevalencies were

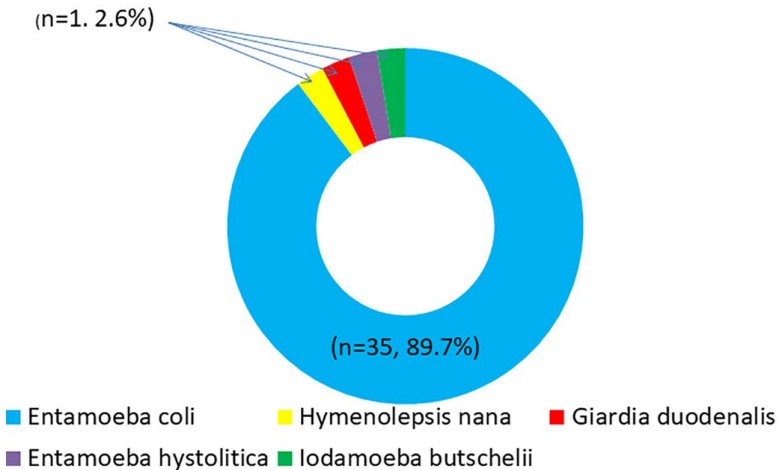

**Fig 2. Doughnut pie chart showing the distribution of intestinal parasitosis.**

conducted in Asia, which is also among the WHO areas endemic for IP. This could be attributable to differences in health care systems, environmental conditions, behavioral practices, and immunological differences among the participants.

The presence of pathogenic intestinal parasitosis in DM immunocompromised patients suggests a risk of increased morbidity and mortality among this population. This is because IP can lead to dehydration, malnutrition, and hospitalization, which can worsen the DM outcomes, such as ketoacidosis and neuropathy. Additionally, the significant burden of IPs among DM patients calls for the need to perform routine screening for these patients, which imposes a cost burden to the health sector, especially in low- and middle-income countries.

Our study found that patients who did not seek medical attention for abdominal symptoms were three [3] times more likely to have intestinal parasites. This is consistent with Sisu et al, who found that patients with no history of visit to a dietician had an increased risk of acquiring IP [12]. Despite our study showing no significant association with known risk factors such as: presence of gastrointestinal symptoms, pet ownership, and rural residence, several other studies showed an association with these factors. IP association with the presence of gastrointestinal symptoms was reported in Ghana [12], and Ethiopia [11], an association with pet ownership was noted in Ethiopia [11], and an association with rural residence was reported in Egypt [20]. Other than these factors, several other factors such as poor educational background, poor sanitation and hygiene, inappropriate latrine use, uncontrolled DM, and presence of complications have also been identified [11–13,20,21]. Pet ownership exposes humans to intestinal parasitosis through direct contact, environmental contamination by the pet's fecal matter, and zoonotic transmission for pets that hunt wild animals. Rural residence stands out as a risk due to an interplay of environmental, behavioral, infrastructural, and socio-economic factors such as open defecation, unprotected wells and rivers, poorly maintained pit latrines, and barefoot walking.

## 5. Limitations

The convenience sampling technique that we used for participant selection poses a risk of bias in the selection and affects the generalizability of our findings. Additionally, our results are not generalizable to the rural population, as more than half of our participants were from the urban population. This urban skew of the sample limits the applicability of our findings to predominantly rural Northern Uganda. We used one stool sample for the diagnosis of intestinal parasites, which reduces sensitivity for detecting intestinal parasitosis, since STHs have intermittent shedding of eggs [22]. This was because our

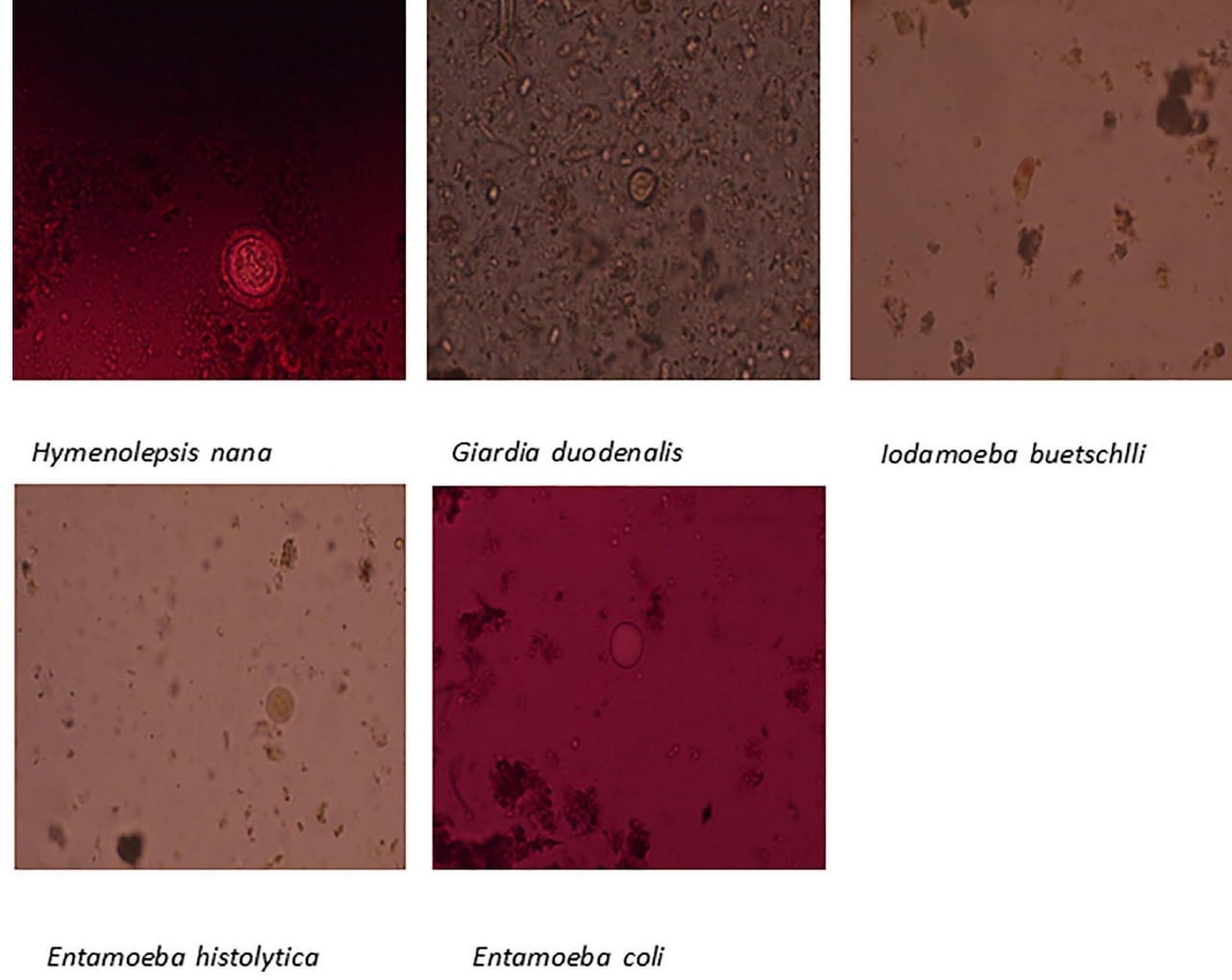

**Fig 3. Images of identified parasites.**

participants were from distant places and could not come to provide stool samples on 3 separate days, and we also had logistical constraints to support the laboratory work for the 3 samples for each patient. We were also not able to perform molecular diagnosis of the intestinal parasites. This was due to logistical constraints and a lack of access.

## 6. Conclusion

The prevalence of intestinal parasitosis was 19.4%. *Entamoeba coli,* a non-pathogenic commensal, constituted the majority, while the other identified parasites were pathogenic: *Hymenolepis nana, Giardia duodenalis, Entamoeba histolytica, and Iodamoeba buetschlii.* Our findings suggest that DM patients remain at risk of acquiring IP, which can impact their quality of life. We recommend routine screening for intestinal parasites and prompt treatment to improve the quality of life of patients with DM, continued health education on sanitation and personal hygiene, and future studies using triplicate stool sampling and molecular methods to enhance detection sensitivity.

**Table 4. Shows analysis of factors associated with intestinal parasitosis among DM patients.**

| Variable | Intestinal Parasites | | cOR (95% CI) | P-value |
|---|---|---|---|---|
| | No (n=162) | Yes (n=39) | | |
| Age | | | 1.0 (0.97–1.03) | 0.928 |
| Fasting Blood Sugar | | | 1.0 (0.98–1.09) | 0.256 |
| Gender | | | | |
| Female | 111 (68.5) | 26 (66.7) | 1 | |
| Male | 51 (31.5) | 13 (33.3) | 0.9 (0.44–1.93) | 0.824 |
| Tribe | | | | |
| Others* | 20 (12.3) | 3 (7.7) | 1 | |
| Acholi | 142 (87.7) | 36 (92.3) | 1.7 (0.48–6.00) | 0.392 |
| Education Level | | | | |
| Primary | 85 (52.5) | 21 (53.8) | 1 | |
| No formal | 32 (19.8) | 10 (25.6) | 1.3 (0.54–2.98) | 0.379 |
| Secondary | 29 (17.9) | 7 (17.9) | 1.0 (0.38–2.54) | |
| Tertiary | 16 (9.9) | 1 (2.6) | 0.3 (0.03–2.02) | |
| Residence | | | | |
| Rural | 78 (48.1) | 18 (46.2) | 1 | |
| Urban | 84 (51.9) | 21 (53.8) | 1.1 (0.54–2.18) | 0.823 |
| Occupation | | | | |
| Civil-servant | 6 (3.7) | 1 (2.6) | 1 | 0.837 |
| Self-employed | 147 (90.7) | 35 (89.7) | 1.4 (0.17–12.25) | |
| Unemployed | 9 (5.6) | 3 (7.7) | 2 (0.17-24.07) | |
| Blood Sugar Control | | | | |
| Controlled | 44 (27.2) | 9 (23.1) | 1 | |
| Uncontrolled | 118 (72.8) | 30 (76.9) | 1.2 (0.55–2.83) | 0.599 |
| Diabetic Complications | | | | |
| No | 22 (13.6) | 5 (12.8) | 1 | 0.900 |
| Yes | 140 (86.4) | 34 (87.2) | 0.9 (0.33–2.65) | |
| Comorbidities | | | | |
| No | 45 (27.8) | 9 (23.1) | 1 | |
| Yes | 117 (72.2) | 30 (76.9) | 1.3 (0.56–2.91) | 0.547 |
| Family History of IP | | | | |
| No | 141 (87.0) | 34 (87.2) | 1 | |
| Yes | 21 (13.0) | 5 (12.8) | 1.0 (0.36–2.88) | 0.981 |
| Pets at home | | | | |
| No | 52 (32.1) | 13 (33.3) | 1 | |
| Yes | 110 (67.9) | 26 (66.7) | 1.1 (0.50–2.22) | 0.883 |
| Water Source | | | | |
| Borehole | 96 (59.3) | 21 (53.8) | 1 | |
| Natural | 32 (19.8) | 10 (25.6) | 1.4 (0.61-3.35) | |
| Piped | 26 (16.0) | 6 (15.4) | 1.1 (0.73–1.55) | 0.755 |
| Mixed | 8 (4.9) | 2 (5.1) | 1.1 (0.22-5.77) | |
| Fecal Disposal | | | | |
| Toilet | 20 (12.3) | 3 (7.7) | 1 | |
| Pit Latrine | 142 (87.7) | 36 (92.3) | 0.6 (0.17–2.10) | 0.392 |
| Current GI Symptoms | | | | |
| Asymptomatic | 45 (27.8) | 10 (25.6) | 1 | |

*(Continued)*

**Table 4.** (Continued)

| Variable | Intestinal Parasites | | cOR (95% CI) | P-value |
|---|---|---|---|---|
| | No (n=162) | Yes (n=39) | | |
| Symptomatic | 117 (72.2) | 29 (74.4) | 1.1 (0.50–2.47) | 0.787 |
| Sought Medical Attention for GI Symptoms *(n=146 symptomatic)* | | | | |
| Yes | 57 (71.3) | 9 (28.1) | 1 | |
| No | 61 (74.4) | 23 (79.3) | 3.0 (1.18–7.53) | 0.014 |

*Other tribes: Langi (13, 6.5%), Alur (5, 2.5%), Lugbara, Muganda, Mukiga, munyankole, munyoro (1 each, 0.5%)

## Supporting information

**S1 File. S1 Questionnaire English.** This is the questionnaire in English.
(PDF)

**S2 File. S2 Dataset Intestinal parasitosis.** This is the dataset for intestinal parasitosis.
(XLSX)

## Acknowledgments

We acknowledge the office of the hospital director of GRRH for allowing us to conduct this study within the hospital. We also thank the staff of GRRH DM clinic and the microbiology laboratory of Prof Emmanuel Odongo-Aginya and Ms Brenda Anena for helping us in analyzing our samples. Above all, we thank the almighty God for protecting us and guiding maneuvering this work

## Author contributions

**Conceptualization:** Hillary Ngolobe, Martin Kugonza, Phillip Nkwatsibwe, Anthony Kato, Donald Otika, Morrish Obol Okello, Daniel S. Ebbs, Felix Bongomin.

**Data curation:** Donald Otika, Morrish Obol Okello, Daniel S. Ebbs, Felix Bongomin.

**Formal analysis:** Hillary Ngolobe, Felix Bongomin.

**Investigation:** Hillary Ngolobe, Martin Kugonza, Phillip Nkwatsibwe, Anthony Kato, Donald Otika, Morrish Obol Okello.

**Methodology:** Hillary Ngolobe, Martin Kugonza, Phillip Nkwatsibwe, Anthony Kato, Donald Otika, Morrish Obol Okello, Felix Bongomin.

**Resources:** Phillip Nkwatsibwe, Anthony Kato.

**Supervision:** Daniel S. Ebbs, Felix Bongomin.

**Validation:** Felix Bongomin.

**Visualization:** Martin Kugonza, Anthony Kato, Donald Otika, Morrish Obol Okello, Daniel S. Ebbs, Felix Bongomin.

**Writing – original draft:** Hillary Ngolobe, Martin Kugonza, Phillip Nkwatsibwe, Anthony Kato, Donald Otika, Morrish Obol Okello, Daniel S. Ebbs, Felix Bongomin.

**Writing – review & editing:** Hillary Ngolobe, Martin Kugonza, Phillip Nkwatsibwe, Anthony Kato, Donald Otika, Morrish Obol Okello, Daniel S. Ebbs, Felix Bongomin.

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
