## [Decision Letter · Decision Letter 0]

31 Oct 2025

Dear Dr. Ngolobe,

Thank you for submitting your manuscript to PLOS ONE. After careful consideration, we feel that it has merit but does not fully meet PLOS ONE’s publication criteria as it currently stands. Therefore, we invite you to submit a revised version of the manuscript that addresses the points raised during the review process.

**ACADEMIC EDITOR:**

We look forward to receiving your revised manuscript.

Kind regards,

Alqeer Aliyo Ali, MSc

Academic Editor

PLOS ONE

Reviewers' comments:

Reviewer's Responses to Questions

**Comments to the Author**

1. Is the manuscript technically sound, and do the data support the conclusions?

Reviewer #1: No

Reviewer #2: Partly

2. Has the statistical analysis been performed appropriately and rigorously?

Reviewer #1: No

Reviewer #2: No

3. Have the authors made all data underlying the findings in their manuscript fully available?

Reviewer #1: No

Reviewer #2: Yes

4. Is the manuscript presented in an intelligible fashion and written in standard English?

Reviewer #1: No

Reviewer #2: Yes

Reviewer #1: 1. Zero STHI prevalence is a major finding but is not sufficiently contextualized within national deworming or public health programs that may explain this result.

2. The title and abstract emphasize STHIs, yet the study found non this creates a mismatch between stated objectives and actual findings.

3. Reframe the study focus: Given the absence of STHIs, consider repositioning the paper to emphasize intestinal parasitosis broadly among DM patients in Uganda.

4. Sample size justification uses Slovin’s formula, which is simplistic and not ideal for prevalence studies; a more robust epidemiological formula (e.g., Cochran’s) would be preferable.

5. Convenience sampling introduces selection bias and limits generalizability; this limitation is underemphasized.

6. Only one stool sample per participant was analyzed, which significantly reduces sensitivity for detecting STHIs due to intermittent egg shedding.

7. Given that no STHIs were detected, how do you reconcile this with Uganda’s reported STHI prevalence of 5–50% in other regions?

8. The methods section lacks detail on how the Kato-Katz and Odongo-Aginya methods were specifically applied (e.g., number of slides, timing, technician training).

9. No molecular diagnostics (e.g., PCR) were used, despite their known higher sensitivity for helminths like Strongyloides stercoralis.

10. The discussion does not adequately explore why STHIs were absent, especially given Uganda’s reported 5–50% STHI prevalence.

11. Entamoeba coli is mischaracterized as “non-pathogenic normal flora”while generally non-pathogenic, its presence indicates fecal-oral contamination and poor sanitation.

12. Include a limitation paragraph explicitly stating that the inability to perform association analyses due to zero STHI cases is a major constraint.

13. Candida albicans is mentioned incidentally but without clinical context was this considered colonization or infection?

14. Statistical analysis of associated factors is missing despite the aim to identify “factors associated with STHIs,” no inferential analysis could be performed due to zero cases, yet this is not explicitly acknowledged.

15. Was seasonality accounted for? The study ran from October 2024 to March 2025 could dry vs. rainy seasons affect STHI transmission?

16. Ethics approval date discrepancy: Approval was granted in 2024, but an amendment is dated 30/05/2025, which is in the future relative to the current date (October 2025). This needs clarification.

17. Inconsistent data availability statement: The submission form states “All relevant data are within the manuscript,” but the manuscript says “Data are available upon reasonable request.”

18. Overstatement of novelty: The claim that data on STHIs in DM patients in Uganda are “scarce” may be true, but the absence of STHIs limits the novelty of findings.

19. No multivariate analysis was possible, yet the abstract implies assessment of “associated factors” this is misleading.

20. Gastrointestinal symptoms (72.6%) are high, yet only 2% were linked to STHIs this disconnect warrants deeper exploration of alternative causes.

21. Pet ownership (67.7%) is noted but not discussed as a potential risk factor for non-STH parasites (e.g., Giardia, Hymenolepis).

22. Urban vs. rural residence is reported, but no analysis compares parasitic infection rates by residence missed opportunity.

23. Fasting blood sugar levels are high (mean 11.9 mmol/L), indicating poor glycemic control, yet no correlation is attempted with parasite presence.

24. Discuss public health implications of finding pathogenic non-STH parasites (e.g., E. histolytica, Giardia) in immunocompromised DM patients.

25. Reference list includes a 2025 citation (Debash et al., 2025) verify if this is an in-press or ahead-of-print article; otherwise, it may be inaccurate.

26. In light of your findings, do you still believe routine STHI screening is warranted for DM patients in this setting, or should screening focus on all intestinal parasites?

Reviewer #2: Thank you editor for providing me this opportunity to review this interesting manuscript. I have completed my review and sending my suggestions and questions for author(s) as follows:

1. Clarify the ethical approval timeline, especially the 2025 amendment date, to avoid confusion or perceived errors.

3. Harmonize the data availability statement across the manuscript and submission form to comply with PLOS ONE policy.

4. Add a subsection in the Discussion on the possible impact of Uganda’s national deworming programs or improved sanitation on STHI reduction.

6. Recommend future studies use triplicate stool sampling and molecular methods to enhance detection sensitivity.

8. Improve methodological transparency: Specify how many Kato-Katz thick smears were prepared per sample and who performed microscopy.

9. Consider subgroup analysis of the 39 parasitized vs. 162 non-parasitized patients to identify risk factors for any intestinal parasite.

10. Revise the title and abstract to reflect the actual findings (e.g., “Absence of Soil-Transmitted Helminths but Presence of Other Intestinal Parasites...”).

Questions for author(s)

11. Why was only one stool sample collected per participant, despite WHO recommendations for multiple samples to improve sensitivity?

12. Were any quality control measures (e.g., duplicate readings, expert validation) implemented during microscopic examination?

13. How was “uncontrolled diabetes” defined was it based solely on fasting blood sugar, or were HbA1c levels also considered?

14. What is the clinical significance of detecting Candida albicans in stool was this considered contamination, colonization, or invasive infection?

15. Did you consider analyzing associations between parasite presence and diabetes complications (e.g., neuropathy, retinopathy)?

16. How generalizable are your findings to rural Ugandan populations, given that over half your sample was urban?

9. Were participants screened for HIV or other immunosuppressive conditions that might influence parasite susceptibility?

**Do you want your identity to be public for this peer review?** For information about this choice, including consent withdrawal, please see our Privacy Policy

Reviewer #1: No

Reviewer #2: No

---

## [Author Response · Author response to Decision Letter 1]

14 Nov 2025

On behalf of the authors’ team for our study “Prevalence and factors associated with Soil-Transmitted Helminths Infections among adults with diabetes mellitus attending a tertiary care facility in Northern Uganda: A hospital-based cross-sectional study”, we sincerely thank the reviewers for the insightful and constructive feedback on our manuscript. The comments we received have significantly strengthened the clarity, rigor, and impact of our work. In the table below, we provided a point-by-point response to each comment we received, explaining the revisions made and indicating where the changes were incorporated in the revised manuscript. We believe these changes address all concerns and enhance the paper’s contribution to the field. We look forward to your kind feedback.

Table showing response to reviewers’ comments

S/No COMMENT RESPONSE PAGE Number (As in track change file)

ACADEMIC EDITOR’S COMMENTS

1 The authors should reframe the title and abstract to reflect the broader focus on intestinal parasitosis The title and abstract have been reframed Page 1 and 2

2. Provide a more robust justification for the sampling strategy We used convenience sampling technique because;

Most of the patients attending the clinic were adults, and a number of them were not willing to provide a stool sample for the study, claiming that it was uncomfortable and that they had no urge to pass stool.

Additionally, we had limited resources to sustain a probability sampling technique, given the limited time we had for data collection.

In light of the above reasons, convenient sampling technique was adapted. Pg 7

3 transparently address methodological limitations such as single stool sampling and lack of molecular diagnostics We collected a single stool sample because we only had the chance to interface with participants as they came for their routine Diabetic care review, which happens at a 1-month interval, and we could therefore not access them easily to provide the 3 separate samples within a 2 to 3 day interval as recommended by WHO.

Also, our participants were from distant places and could not come to provide stool samples on 3 separate days, due to transportation constraints.

We also had logistical constraints to support the laboratory work for the 3 samples for each patient Pg 24

4. Additionally, the ethical approval timeline We received initial approval on 24/09/2024 and started data collection in October 2024. We then made amendments, and the amendment approval was given on 30/05/2025 Pg 8

5. Data availability statement must be clarified and harmonized to meet journal standards We have addressed this by emphasizing that; All relevant data are within the manuscript and its supporting information files. Data are available upon reasonable request from the first author Pg 27

REVIEWER 1 COMMENTS

1. Zero STHI prevalence is a major finding but is not sufficiently contextualized within national deworming or public health programs that may explain this result. We have considered intestinal parasitosis instead of STHIs, as guided by the academic editor.

This change has been embraced throughout the manuscript.

2. The title and abstract emphasize STHIs, yet the study found non this creates a mismatch between stated objectives and actual findings. The title and abstract have been modified to emphasize intestinal parasitosis, and the objectives and findings have also been restructured in line with intestinal parasitosis Title page, Pg 1,2,

3. Reframe the study focus: Given the absence of STHIs, consider repositioning the paper to emphasize intestinal parasitosis broadly among DM patients in Uganda. We have reframed the study to focus on intestinal parasitosis Throughout the manuscript

4. Sample size justification uses Slovin’s formula, which is simplistic and not ideal for prevalence studies; a more robust epidemiological formula (e.g., Cochran’s) would be preferable. We've adjusted to use the Cochran formula, which gives us a sample size of 267 participants and a response rate of 75.3% (n=201/267) Pg 7

5. Convenience sampling introduces selection bias and limits generalizability; this limitation is underemphasized. We have included this limitation in our limitations section Pg 24

6. Only one stool sample per participant was analyzed, which significantly reduces sensitivity for detecting STHIs due to intermittent egg shedding. This was because we only had the chance to interface with participants as they came for their routine Diabetic care review, which happens at a 1-month interval, and we could therefore not access them easily to provide the 3 separate samples within a 2 to 3 day interval as recommended by WHO.

Also, our participants were from distant places and could not come to provide stool samples on 3 separate days, due to transportation constraints.

We also had logistical constraints to support the laboratory work for the 3 samples for each patient Pg 24

7. Given that no STHIs were detected, how do you reconcile this with Uganda’s reported STHI prevalence of 5–50% in other regions? We have reframed our manuscript to focus on intestinal parasites rather than STHIs.

With the reframing, our study now emphasizes the presence of intestinal parasitosis, at a prevalence of 19.4% unlike the initial absence of STHIs. Pg 16

8. The methods section lacks detail on how the Kato-Katz and Odongo-Aginya methods were specifically applied (e.g., number of slides, timing, technician training). This has been addressed. Two thick smears were prepared, and analysis was done within a 4-hour interval from the time of sample collection. The stool analysis was performed by 1 laboratory technician and 1 professor of parasitology. Pg 7

9. No molecular diagnostics (e.g., PCR) were used, despite their known higher sensitivity for helminths like Strongyloides stercoralis. This was not possible due to limited logistics. No PCR tests were being done in the laboratory that analyzed our samples, and additionally, we had limited funds to support analysis elsewhere. Pg 24

10. The discussion does not adequately explore why STHIs were absent, especially given Uganda’s reported 5–50% STHI prevalence. The discussion has been reframed to focus on intestinal parasitosis and not STHIs, since they were absent Pg 19-23

11. Entamoeba coli is mischaracterized as “non-pathogenic normal flora,” while generally non-pathogenic, its presence indicates fecal-oral contamination and poor sanitation. We have re-characterized it under pathogenic parasites and discussed the need to embrace the WASH strategy to mitigate its transmission, since its presence indicates ongoing fecal-oral contamination. Abstract and Pg 19

12. Include a limitation paragraph explicitly stating that the inability to perform association analyses due to zero STHI cases is a major constraint. Since we’ve reframed our manuscript to focus on intestinal parasitosis, and we found intestinal parasitosis in 19.4% of the participants, we have performed association analyses Pg 17-19

13. Candida albicans is mentioned incidentally but without a clinical context. Was this considered colonization or infection? It was considered a colonization

14. Statistical analysis of associated factors is missing despite the aim to identify “factors associated with STHIs,” no inferential analysis could be performed due to zero cases, yet this is not explicitly acknowledged. We’ve reframed our manuscript to focus on intestinal parasitosis, and since we found intestinal parasitosis, inferential analysis has been performed Pg 17-19

15. Was seasonality accounted for? The study ran from October 2024 to March 2025. Could dry vs. rainy seasons affect STHI transmission? Yes, seasons affect STHI transmission, with the transmission being higher during rainy seasons due to increased contamination from running water and latrine overflows, and favorable conditions of breeding for the vectors.

The wet season in our study area ranges from April to November, and the dry season ranges from November to March.

However, we couldn't halt data collection until the wet season begins, as we would not meet the timeline that had been given to us to conduct the study.

16. Ethics approval date discrepancy: Approval was granted in 2024, but an amendment is dated 30/05/2025, which is in the future relative to the current date (October 2025). This needs clarification. We received initial approval on 29/09/2024 and started data collection in October 2024. We then made amendments, and the amendment approval was given on 30/05/2025. Pg 8

17. Inconsistent data availability statement: The submission form states “All relevant data are within the manuscript,” but the manuscript says “Data are available upon reasonable request. This has been corrected, emphasizing that: All relevant data are within the manuscript and its supporting information files. Data are available upon reasonable request from the first author Pg 27

18. Overstatement of novelty: The claim that data on STHIs in DM patients in Uganda are “scarce” may be true, but the absence of STHIs limits the novelty of findings To address this, our manuscript has been reframed to focus on intestinal parasitosis rather than STHIs Throughout the manuscript

19. No multivariate analysis was possible, yet the abstract implies assessment of “associated factors.” this is misleading. With a focus on intestinal parasitosis, factor analysis was possible, and it has been performed Pg 17-19

20. Gastrointestinal symptoms (72.6%) are high, yet only 2% were linked to STHIs this disconnect warrants deeper exploration of alternative causes. We agree to this.

21. Pet ownership (67.7%) is noted but not discussed as a potential risk factor for non-STH parasites (e.g., Giardia, Hymenolepis). We have discussed pet ownership as a risk factor for intestinal parasitosis. Pg 22

22. Urban vs. rural residence is reported, but no analysis compares parasitic infection rates by residence, a missed opportunity. We have performed an analysis to compare parasitic infection by residence and found no association Pg 17-19

23. Fasting blood sugar levels are high (mean 11.9 mmol/L), indicating poor glycemic control, yet no correlation is attempted with parasite presence. We have performed a correlation for Fasting Blood sugar with intestinal parasites, though it wasn’t found to be significant (p=0.256) Pg 17

24. Discuss public health implications of finding pathogenic non-STH parasites (e.g., E. histolytica, Giardia) in immunocompromised DM patients. This has been discussed, and we have emphasized the risk of worsening morbidity and mortality in this population. Additionally, implementing screening for these patients generates a direct cost impact on the health sector. Pg 21

25. Reference list includes a 2025 citation (Debash et al., 2025). Verify if this is an in-press or ahead-of-print article; otherwise, it may be inaccurate. We’ve verified and found that it’s a valid article, published in the BMC Infectious Diseases journal (https://doi.org/10.1186/s12879-025-10441-4)

26. In light of your findings, do you still believe routine STHI screening is warranted for DM patients in this setting, or should screening focus on all intestinal parasites? Screening should focus on all intestinal parasites and not only STHI as previously thought Pg 25

REVIEWER 2 COMMENTS

1. Clarify the ethical approval timeline, especially the 2025 amendment date, to avoid confusion or perceived errors. We received initial approval on 29/09/2024 and started data collection in October 2024. We then made amendments, and the amendment approval was given on 30/05/2025 Pg 8

2. Harmonize the data availability statement across the manuscript and submission form to comply with PLOS ONE policy. We have harmonized this emphasizing that; All relevant data are within the manuscript and its supporting information files. Data are available upon reasonable request from the first author Pg 27

3. Add a subsection in the Discussion on the possible impact of Uganda’s national deworming programs or improved sanitation on STHI reduction. With a focus on intestinal parasitosis, our study revealed evidence of poor sanitation (ongoing fecal oral contamination) by the presence of Entamoeba coli. We therefore recommended enhancement of the WASH strategy and screening as preventive strategies to combat this ongoing spread. Pg 19, 25

4. Recommend future studies use triplicate stool sampling and molecular methods to enhance detection sensitivity. This recommendation has been included Pg 25

5. Improve methodological transparency: Specify how many Kato-Katz thick smears were prepared per sample and who performed microscopy. This has been addressed.

Two (2) smears were prepared by a laboratory technician and a parasitology professor. Pg 7

6. Consider subgroup analysis of the 39 parasitized vs. 162 non-parasitized patients to identify risk factors for any intestinal parasite. We have performed this subgroup analysis. Pg 17-19

7. Revise the title and abstract to reflect the actual findings (e.g., “Absence of Soil-Transmitted Helminths but Presence of Other Intestinal Parasites...”). The title and abstract have been revised Title page, Pg 1,2

8. Why was only one stool sample collected per participant, despite WHO recommendations for multiple samples to improve sensitivity? We obtained a single stool sample because patients come to the clinic every 1 month and could not meet the 3-day WHO-recommended interval between stool sample collection.

Additionally, most of our participants resided in locations distant from our study site and couldn’t come on 3 separate days to provide samples.

Furthermore, there were limited funds to facilitate the laboratory technician to analyze the 3 samples.

We did not have access to any molecular diagnostics, since they weren’t available in the laboratory at our study site, and there were limited funds to transport and facilitate molecular diagnosis elsewhere.

9. Were any quality control measures (e.g., duplicate readings, expert validation) implemented during microscopic examination? Quality control measures were implemented. Stool samples that had similar identification numbers were not considered for analysis to avoid duplicates. Every parasite that was not clear to the laboratory technician was proofread and confirmed by the professor of parasitology.

10. How was “uncontrolled diabetes” defined was it based solely on fasting blood sugar, or were HbA1c levels also considered? Uncontrolled diabetes was defined based on fasting blood sugar, because we could not perform HbA1c, since it was absent at the study site, and needed funds to be done elsewhere, which we didn’t have, as the study was not funded.

11. What is the clinical significance of detecting Candida albicans in stool? Was this considered contamination, colonization, or invasive infection? There is no clinical significance of finding Candida in stool. We considered it a contamination, and for clarity, we have removed Candida from our results. All through the manuscript.

12. Did you consider analyzing associations between parasite presence and diabetes complications (e.g., neuropathy, retinopathy)? Analysis of associations has been done, and we found no association of intestinal parasites with DM complications (p=0.900) Pg 17-19

13. How generalizable are your findings to rural Ugandan populations, given that over half your sample was urban? Our findings are not generalizable to the rural population. We have included this in our limitations

14. Were participants screened for HIV or other immunosuppressive conditions that might influence parasite susceptibility? No, we did not screen the participants for HIV or other immunosuppressive conditions

---

## [Decision Letter · Decision Letter 1]

25 Nov 2025

Dear Dr. Ngolobe,

Thank you for submitting your manuscript to PLOS ONE. After careful consideration, we feel that it has merit but does not fully meet PLOS ONE’s publication criteria as it currently stands. Therefore, we invite you to submit a revised version of the manuscript that addresses the points raised during the review process.

<small>The authors have been highly responsive and have successfully reframed their paper, which is a significant improvement. The requested changes are not major but are essential for the manuscript's scientific accuracy, clarity, and focus. The authors are commended for their thorough revisions and for refocusing the manuscript on intestinal parasitosis, which has significantly strengthened the paper. However, minor revisions are required to address lingering issues related to the characterization of *Entamoeba coli* , the focus of the discussion, and minor inconsistencies in dates and grammar. Please carefully review the points listed above and submit a final version for editorial consideration. Here are the key issues that should be addressed in a final round of revisions:</small>

<h4 style="font-variant-numeric: normal; font-variant-east-asian: normal; font-variant-alternates: normal; font-size-adjust: none; font-kerning: auto; font-optical-sizing: auto; font-feature-settings: normal; font-variation-settings: normal; font-variant-position: normal; font-variant-emoji: normal; font-stretch: normal; font-size: 16px; line-height: 28px; font-family: quote-cjk-patch, Inter, system-ui, -apple-system, BlinkMacSystemFont, "Segoe UI", Roboto, Oxygen, Ubuntu, Cantarell, "Open Sans", "Helvetica Neue", sans-serif; margin: 16px 0px 8px; color: rgb(15, 17, 21); background-color: rgb(255, 255, 255);"><small>1. Abstract - Grammar and Clarity</small></h4>

<small>The abstract contains several grammatical errors and awkward phrases that affect readability.</small><small>Example: "Descriptive and inferential statistics were performed to assess the prevalence..." → This is redundant. Suggest: "Descriptive statistics and logistic regression were used to determine the prevalence and identify factors associated with IP."</small><small>Recommendation: A thorough proofread of the abstract is needed to improve flow and correct minor grammatical errors.</small>

<h4 style="font-variant-numeric: normal; font-variant-east-asian: normal; font-variant-alternates: normal; font-size-adjust: none; font-kerning: auto; font-optical-sizing: auto; font-feature-settings: normal; font-variation-settings: normal; font-variant-position: normal; font-variant-emoji: normal; font-stretch: normal; font-size: 16px; line-height: 28px; font-family: quote-cjk-patch, Inter, system-ui, -apple-system, BlinkMacSystemFont, "Segoe UI", Roboto, Oxygen, Ubuntu, Cantarell, "Open Sans", "Helvetica Neue", sans-serif; margin: 16px 0px 8px; color: rgb(15, 17, 21); background-color: rgb(255, 255, 255);"><small>2. Characterization of *Entamoeba coli* </small></h4>

<small>The authors have correctly noted that the presence of *E. coli*  indicates fecal-oral contamination. However, labeling it as a "pathogenic" parasite in the Abstract (line 25) and Results (line 153) is scientifically inaccurate.</small><small>*Entamoeba coli*  is widely recognized as a non-pathogenic commensal. Its significance lies in its role as a marker of poor sanitation, not as a cause of disease.</small><small>Recommendation: Consistently refer to *E. coli*  as a "non-pathogenic intestinal protozoan" or "commensal" throughout the manuscript. The public health message remains strong: its presence indicates a risk of exposure to truly pathogenic parasites.</small>

<h4 style="font-variant-numeric: normal; font-variant-east-asian: normal; font-variant-alternates: normal; font-size-adjust: none; font-kerning: auto; font-optical-sizing: auto; font-feature-settings: normal; font-variation-settings: normal; font-variant-position: normal; font-variant-emoji: normal; font-stretch: normal; font-size: 16px; line-height: 28px; font-family: quote-cjk-patch, Inter, system-ui, -apple-system, BlinkMacSystemFont, "Segoe UI", Roboto, Oxygen, Ubuntu, Cantarell, "Open Sans", "Helvetica Neue", sans-serif; margin: 16px 0px 8px; color: rgb(15, 17, 21); background-color: rgb(255, 255, 255);"><small>3. Discussion Section - Lingering STH Content</small></h4>

<small>The Discussion still contains a long paragraph (pages 58-59, lines 242-261) discussing factors associated with STH and *Strongyloides stercoralis* , which were not found in this study.</small><small>This section feels out of place and dilutes the new focus on the intestinal parasites that *were*  actually found.</small><small>Recommendation: Remove or significantly shorten this paragraph. The discussion should focus on interpreting the findings of *this*  study (e.g., the high prevalence of *E. coli* , the single significant risk factor, the implications of finding *Giardia*  and *E. histolytica*  in a diabetic population).</small>

<h4 style="font-variant-numeric: normal; font-variant-east-asian: normal; font-variant-alternates: normal; font-size-adjust: none; font-kerning: auto; font-optical-sizing: auto; font-feature-settings: normal; font-variation-settings: normal; font-variant-position: normal; font-variant-emoji: normal; font-stretch: normal; font-size: 16px; line-height: 28px; font-family: quote-cjk-patch, Inter, system-ui, -apple-system, BlinkMacSystemFont, "Segoe UI", Roboto, Oxygen, Ubuntu, Cantarell, "Open Sans", "Helvetica Neue", sans-serif; margin: 16px 0px 8px; color: rgb(15, 17, 21); background-color: rgb(255, 255, 255);"><small>4. Ethics Statement - Date Inconsistency</small></h4>

<small>In the manuscript (page 18, line 101) and the ethics statement (page 7), the initial approval date is listed as "29/09/2025". This is a clear typo, as data collection started in October 2024.</small><small>The response to reviewers correctly states it as "29/09/2024".</small><small>Recommendation: Correct this date to "29/09/2024" in the manuscript's methods section and ethics statement to avoid confusion and maintain credibility.</small>

<h4 style="font-variant-numeric: normal; font-variant-east-asian: normal; font-variant-alternates: normal; font-size-adjust: none; font-kerning: auto; font-optical-sizing: auto; font-feature-settings: normal; font-variation-settings: normal; font-variant-position: normal; font-variant-emoji: normal; font-stretch: normal; font-size: 16px; line-height: 28px; font-family: quote-cjk-patch, Inter, system-ui, -apple-system, BlinkMacSystemFont, "Segoe UI", Roboto, Oxygen, Ubuntu, Cantarell, "Open Sans", "Helvetica Neue", sans-serif; margin: 16px 0px 8px; color: rgb(15, 17, 21); background-color: rgb(255, 255, 255);"><small>5. Limitations Section - Emphasis on Generalizability</small></h4>

<small>The authors rightly note that their findings are not generalizable to the rural population. This is a crucial limitation.</small><small>Recommendation: Consider strengthening this point by explicitly stating that the urban skew of the sample limits the applicability of the findings to predominantly rural Northern Uganda.</small>

We look forward to receiving your revised manuscript.

Kind regards,

Alqeer Aliyo Ali, MSc

Academic Editor

PLOS ONE

**Journal Requirements:**

Reviewers' comments:

Reviewer's Responses to Questions

**Comments to the Author**

Reviewer #1: All comments have been addressed

Reviewer #2: All comments have been addressed

2. Is the manuscript technically sound, and do the data support the conclusions?

Reviewer #1: Yes

Reviewer #2: Yes

3. Has the statistical analysis been performed appropriately and rigorously?

Reviewer #1: Yes

Reviewer #2: Yes

4. Have the authors made all data underlying the findings in their manuscript fully available?

Reviewer #1: Yes

Reviewer #2: Yes

5. Is the manuscript presented in an intelligible fashion and written in standard English?

Reviewer #1: Yes

Reviewer #2: Yes

Reviewer #1: The author(s) have revises the manuscript and responded my queries accordingly. Therefore, the manuscript now acceptable.

Reviewer #2: I satisfied with the author corrections as well as the changes he/her made. I recommend the article should be considered for publication.

**Do you want your identity to be public for this peer review?** For information about this choice, including consent withdrawal, please see our Privacy Policy

Reviewer #1: No

Reviewer #2: No

---

## [Author Response · Author response to Decision Letter 2]

26 Nov 2025

RESPONSE TO REVIEWERS II

On behalf of the authors’ team for our study “Prevalence and factors associated with Intestinal parasitosis among adults with diabetes mellitus attending a tertiary care facility in Northern Uganda: A hospital-based cross-sectional study”, we sincerely thank the reviewers for the additional feedback and comments on our manuscript. The comments we received have significantly improved the clarity, rigor, and scientific impact of our work. In the table below, we provided a point-by-point response to each comment we received, explaining the revisions made and indicating where the changes were incorporated in the revised manuscript. We believe these changes address all concerns and enhance the paper’s contribution to the field. We look forward to your kind feedback.

S/No Comment Response Page in mark-up copy

1 The abstract contains several grammatical errors and awkward phrases that affect readability. This has been addressed Page 1 and 2

2. Consistently refer to E. coli as a "non-pathogenic intestinal protozoan" or "commensal" throughout the manuscript. We have corrected this Line 35 (Page 2),

Line 180 (Page 14)

Line 206 (Page 17)

Line 265 (Page 20)

3. The Discussion still contains a long paragraph (pages 58-59, lines 242-261) discussing factors associated with STH and Strongyloides stercoralis, which were not found in this study This has been rectified.

4. In the manuscript (page 18, line 101) and the ethics statement (page 7), the initial approval date is listed as "29/09/2025". This is a clear typo, as data collection started in October 2024. We have corrected this Line 122 (Page 6) in the manuscript and also in the ethical statement.

5. Consider strengthening this point by explicitly stating that the urban skew of the sample limits the applicability of the findings to predominantly rural Northern Uganda. We have addressed this Line 256 (Page 20)

---

## [Editor Report · Decision Letter 2]

28 Nov 2025

Prevalence and factors associated with Intestinal parasitosis among adults with diabetes mellitus attending a tertiary care facility in Northern Uganda: A hospital-based cross-sectional study

PONE-D-25-53019R2

Dear Dr. Ngolobe,

We’re pleased to inform you that your manuscript has been judged scientifically suitable for publication and will be formally accepted for publication once it meets all outstanding technical requirements.

Kind regards,

Alqeer Aliyo Ali, MSc

Academic Editor

PLOS ONE
---

## [Editor Report · Acceptance letter]

PONE-D-25-53019R2

PLOS One

Dear Dr. Ngolobe,

I'm pleased to inform you that your manuscript has been deemed suitable for publication in PLOS One. Congratulations! Your manuscript is now being handed over to our production team.

Kind regards,

on behalf of

Mr. Alqeer Aliyo Ali

Academic Editor

PLOS One